# Non-communicable disease and mental health care during the COVID-19 pandemic in South Africa: Perspectives from selected healthcare professionals and patients

**Nasheeta Peer**[1,2]*, **Tshephang Mashiane**[1], **Michel Oris**[3,4], **Kibachio Mwangi**[5], **Sugitha Sureshkumar**[3], **Andre-Pascal Kengne**[1,2]

**1** Non-communicable Diseases Research Unit, South African Medical Research Council, Durban and Cape Town, South Africa, **2** Department of Medicine, University of Cape Town, Cape Town, South Africa, **3** Institute of Global Health (IGH), University of Geneva (UNIGE), Geneva, Switzerland, **4** Centre LIVES, University of Geneva, Geneva, Switzerland, **5** World Health Organization (WHO) Country Office, Pretoria, Tshwane, South Africa

* nasheeta.peer@mrc.ac.za

## Abstract

### Aim

South Africa has a high burden of non-communicable diseases (NCDs) and experienced a high COVID-19 caseload, while the healthcare system was already overstretched. The aim of this study was to examine the perceptions and experiences of 1) healthcare professionals (HPs) of COVID-19 and NCD management, and the preparedness of health systems to provide NCD care, and 2) people living with NCDs (PLWNCDs) on the care they received, their mental health status and the availability of health information during the COVID-19 pandemic in South Africa.

### Methods

We recruited a convenience sample of 1) HPs who worked in healthcare management, i.e., public health officials and healthcare workers who provided care and engaged with patients in any healthcare capacity, and 2) PLWNCDs with ≥1 NCD. Questionnaires comprised quantitative and some open-ended follow-up questions. SPSS was used to analyse the quantitative data, and content analysis with inductive reasoning was used to evaluate the open-ended questions.

### Results

This cross-sectional study comprised 31 HPs and 79 PLWNCDs. The provision of COVID-19 care was perceived to be adequate by HPs while NCD care was poor with disruptions of services, including for emergency and specialised NCDs care. Strategies to care for non-COVID-19 illnesses during the pandemic were lacking. This would have serious long-term consequences for PLWNCDs and the healthcare system. Perceptions of inadequate NCD care were repeated by PLWNCDs; 49% felt they received 'very little' or no adequate

**Data availability statement:** All relevant data are within the manuscript and its Supporting Information files.

**Funding:** The author(s) received no specific funding for this work.

**Competing interests:** The authors have declared that no competing interests exist.

care and 18% had a scheduled appointment cancelled. Further, many PLWNCDs (52%) felt anxious, lonely or frightened during the pandemic and 15% felt their mental health had deteriorated, but only a small proportion sought medical attention. The utility of digital health was positively perceived by both HPs and PLWNCDs and could contribute to better health provision during crises.

## Conclusions

Policies are needed to ensure that NCD care will not be neglected during future crises and to encourage PLWNCDs to access healthcare services timeously during such periods. Potential strategies may utilise digital health apps to improve care and to address the mental healthcare needs of PLWNCDs.

## Introduction

Healthcare systems in low- and middle-income countries (LMICs), including South Africa are generally overburdened and underfunded with care directed to acute conditions or chronic infectious diseases such as HIV/AIDS and tuberculosis [1]. Consequently, non-communicable disease (NCD) care for hypertension, diabetes, and other NCDs have not been prioritised with healthcare stakeholders perceiving these conditions to be less important [1]. NCD care has been deemed less important despite the call by the Global Action Plan on NCDs to strengthen healthcare systems for the effective management of NCDs [2] and the Political Declaration of the High-Level Meeting of the General Assembly on the Prevention and Control of NCDs in 2011 where South Africa was at the forefront on drafting and ratifying the declaration [3]. The unmet need for NCD care has serious consequences for the individual at the health, societal and economic levels, particularly when it affects the working age population. NCDs contribute to disability and premature death, with the majority occurring in LMICs [4].

The rapid spread of the 2019 Corona Virus Disease (COVID-19) across the globe brought these healthcare disparities into sharp focus by aggravating pre-existing healthcare weaknesses in capacity and capabilities [5–8]. There was a sudden and major disruption in the provision of healthcare services for NCDs with all key resources directed at managing the COVID-19 pandemic [4,9,10]. Healthcare systems were tested to their limits when faced with this new unprecedented public health challenge [9]. Many countries focused their resources on containing the COVID-19 pandemic at the expense of continued provision of healthcare services for other conditions, thereby compromising the care of non-pandemic related conditions. Healthcare providers were repurposed from routine care to bolster the public health efforts to halt and reverse the burden of COVID-19 further compromising NCD care. Notably, this occurred even though people living with NCDs (PLWNCDs) were found to be more vulnerable to the COVID-19 virus and had worse outcomes [4,11]. The COVID-19 disease burden, the available resources (human, financial, infrastructure, etc.), the functionality of health systems, and political commitment varied dramatically across countries [10]. Moreover, the reaction of countries to the pandemic in terms of policies and responses, and their redirection of health systems and resources for COVID-19 care differed widely [9]. This would, therefore, differentially impact NCD care in countries worldwide.

It is important to understand the shortcomings of each country's healthcare system during COVID-19, the impact it had on both healthcare providers and PLWNCDs and pave the way forward to better integrate and improve healthcare delivery for NCDs during and after the pandemic. This is highly relevant for South Africa which reported the highest cumulative

COVID-19 cases in Africa; unsurprisingly, this overwhelmed the country's healthcare system [8,12]. South Africa underwent a prolonged national lockdown (>2 years) because of the pandemic which, together with disrupting healthcare systems and affecting patients' access to care, limited social gatherings and interactions which had a negative impact on people's livelihoods, and their psyche and mental health [7,13]. Therefore, the aim of this study was to examine the perceptions and experiences of 1) healthcare professionals (HP) of COVID-19 and NCD management, and the preparedness of health systems to provide NCD care, and 2) PLWNCDs on the care they received, their mental health status and the availability of health information and services during the COVID-19 pandemic in South Africa.

## Materials and methods

### Study population and sampling procedure

This was a collaborative study between scientists from the Non-communicable Diseases Research Unit of the South African Medical Research Council (SAMRC) and the Institute of Global Health, University of Geneva. Study participants comprised 1) HPs who worked in healthcare management, i.e., public health officials (PHO), or provided care and engaged with patients in any healthcare capacity, i.e., healthcare workers (HW), and 2) people with ≥1 NCD who had been diagnosed and received treatment prior to the COVID-19 pandemic. Participants >18 years of age, working or receiving care in any public or private healthcare facility in South Africa were eligible for inclusion in the study. A minimal of 10 participants per group (PHOs, HWs and PLWNCDs) was targeted.

A convenience sample of participants was recruited in this cross-sectional study via HP and patient organisations, and snowballing sampling via own networks using email, telephone, and social media posts to raise awareness of this study and recruit participants. The study questionnaire was developed through consultation with the COVID-19 ELEPHANT study group. The questions were tested for acceptability by two independent groups with diverse professional backgrounds, which further informed the development of the questionnaire. Questionnaires for the two categories of HPs were similar but also included specific care related questions for HWs. The self-administered questionnaires comprised quantitative and some open-ended follow-up questions. HPs first provided written informed consent electronically; thereafter, they completed a password protected questionnaire and returned this via email.

PLWNCDs completed quantitative questionnaires that were administered according to their preference, i.e., interviewer administered in-person questionnaires was the main preference followed by interviewer administered telephonic interviews, and a few participants requested self-administered emailed questionnaires. All participants provided informed consent before the questionnaires were administered.

Ethical approval was provided by the SAMRC Ethics Committee (protocol ID EC031–7/2021). The study was conducted in accordance with principles of the International Declaration of Helsinki, 2013.

### Data collection

Data collection was conducted over a 10-month period from October 2021 to July 2022 even though the targeted minimum 10 participants per group had not been obtained for PHOs; however, participant numbers for HWs and PLWNCDs had exceeded the minimum required. The fieldwork was terminated at this stage because the COVID-19 pandemic had passed its peak and all COVID-19 restrictions (e.g., wearing of face mask, social distancing, temperature checks, curfews, etc.) had been lifted in June 2022 in South Africa. The study team felt that any interviews conducted after the lifting of restrictions may be prone to greater recall bias [14,15].

Relevant self-reported sociodemographic data (age, gender, healthcare sector, province, etc.) were collected from all participants, as well as a brief medical history of comorbidities from PLWNCDs. Data collected from HPs related to NCD and COVID-19 management including personal experiences and fears, perceptions of their patients' behaviours and mental health, utility of digital health, ethical issues pertaining to the pandemic, and lessons learnt for the future. PLWNCDs were asked about their NCD care, mental health, and the availability of information during the COVID-19 pandemic.

## Analyses

Data analyses, conducted using SPSS version 28.0, comprised descriptive statistics to summarise the quantitative data. Frequency (n) and percentage (%) were used for categorical variables. Continuous variables were not normally distributed and were thus presented as medians and 25th-75th percentiles.

The method used to evaluate the open-ended questions was content analysis [16]. Valid inferences were systematically and objectively drawn from the HPs' written text responses to these open-ended questions. Manifest analysis was used in this study with frequent referrals to the original participant text thereby retaining the original meanings and contexts. Inductive reasoning, where conclusions are developed from the data collected, was used. Each open-ended question for HPs was examined for common perceptions among participants, which were then summarised. Some texts have been quoted verbatim, with the permission of the HP. All data presented have been anonymised.

## Results

1) **Healthcare professionals: public health officials and healthcare workers**

### Participant characteristics

The 31 HPs interviewed comprised six PHOs (hospital, nursing and pharmacy managers) and 25 HWs (Fig 1). HWs included 13 doctors and 12 other HWs consisting of nurses, pharmacists, dieticians, an optometrist, a physiotherapist and an occupational therapist. The overall median age (25th–75th percentiles) of the HPs was 31.0 (25.0–37.0) years and their median number of years of work experience was 9.0 (4.0–9.0) years (S1 Fig). Of the six men and 25 women interviewed, 80% personally worked with COVID-19 patients for a duration of 2 months to 2 years. All PHOs worked in the private healthcare sector while 13 HWs worked in the public and 12 in the private healthcare sector.

### Provision of COVID-19 care during the pandemic

Many HPs (48%) felt that planning was poor with regards to contingency plans to shield the healthcare system in case of an external disruption like COVID-19. This extended to a lack of protocols and guidelines to concentrate services in high-volume, high-acuity settings with 24-hour care (36%) (Table 1). Consequently, most HWs (60%) felt that patients who would have received a cubicle/isolated space in a department/ward were placed in shared space/ward (Table 2). The majority of HPs (68%) felt that the extent to which the government and policymakers in the country or hospital managers at their health facility were aware of the risk to frontline workers was 'quite a bit' or they were completely aware, while 29% felt they were a little or not at all aware (Table 1). About half the HWs (56%) felt that dedicated personnel were often rotated so as not to exhaust them.

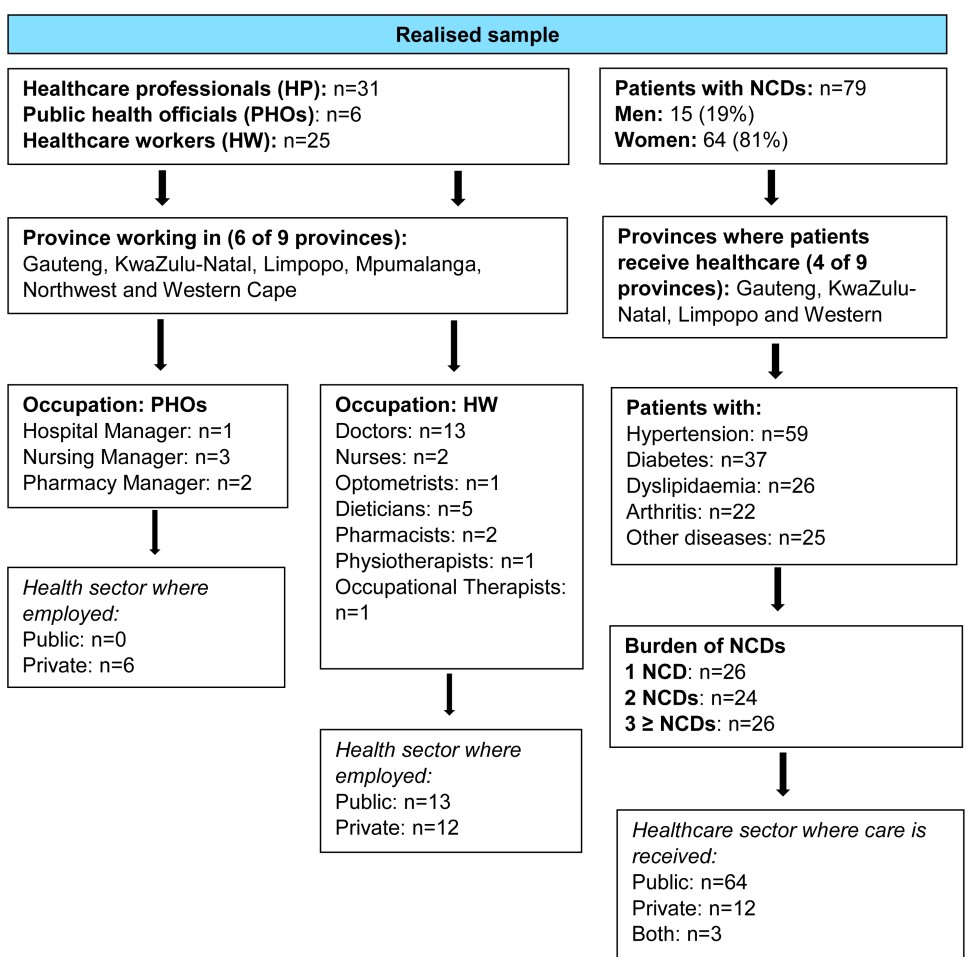

**Fig 1. Characteristics of study population.**

In general, the provision of COVID-19 care was perceived to be adequate with the creation of multidisciplinary teams for care (68%) and HWs received adequate personal protection equipment (PPE) dependent on their workplace setting (77%) (Table 1). Most HWs (80%) felt that the extent to which they received adequate personal protection was either 'quite a bit' or completely (Table 2). However, masks frequently had to be reused with limited supplies, and shoe covers were not provided. Most HWs (76%) felt that they were well perceived by patients whilst wearing PPE. About half the HWs (52%) did not feel detached from patients whilst wearing PPE.

## Provision of NCD care during the pandemic

Many HPs felt that the level of NCD care in South Africa prior to the COVID-19 pandemic was good (58%) but almost a quarter (23%) felt it was 'not good' (Table 1). However, during the pandemic, 78% of HPs perceived the severity of NCD service disruption (including elective surgery, follow-up clinics, routine care) to be 'very often' or 'severe'.

HWs reported that in hospitals, some wards that usually treated NCD patients were temporarily closed to cater for COVID-19 patients with staff reallocated to acute care; emergency services, dialysis and acute haematology and oncology care were still available. Elective surgery such as renal transplants, cardiac bypass surgery and treatment for diabetes

**Table 1. Healthcare professionals' experiences of patient care, personal perceptions, and ethical concerns during the COVID-19 pandemic\*.**

| Provision of COVID-19 care during the pandemic: n (%) | Completely | Quite a bit | A little bit/ not at all |
|---|---|---|---|
| Felt hospital managers at the health facility or government and policy makers in the country were aware of the risk to frontline workers (n=30) | 4 (12.9) | 17 (54.8) | 9 (29.0) |
| | **Yes** | **No** | **Unsure** |
| Contingency plans existed to shield the healthcare system in case of external disruption like COVID-19 | 6 (19.4) | 15 (48.4) | 10 (32.3) |
| Chronic care management was redirected to focus on maintaining supply chains for COVID-19 | 9 (29.0) | 11 (35.5) | 11 (35.5) |
| Multidisciplinary teams were created intentionally for working in COVID-19 wards (n=28) | 21 (67.7) | 7 (22.6) | N/A |
| In general, healthcare workers received adequate personal protection equipment depending on their workplace setting (n=29) | 24 (77.4) | 5 (16.1) | N/A |
| *Provision of non-communicable disease (NCD) care during the pandemic: n (%)* | **Good** | **Satisfactory** | **Not good** |
| Level of NCD care in South Africa prior to COVID-19 pandemic (n=30): | 18 (58.1) | 5 (16.1) | 7 (22.6) |
| | **Severe** | **Very often** | **Occasionally/ never** |
| Severity of NCD service disruption (including elective surgery, follow-up clinics, routine care) during the pandemic | 11 (35.5) | 13 (41.9) | 7 (22.6) |
| | **Yes** | **No** | **Unsure** |
| Afflicted by health system's inability to continue caring for patients with NCDs | 24 (77.4) | 7 (22.6) | N/A |
| Worried that NCD patients who had not been followed up due to the crisis will have worsened outcomes | 14 (45.2) | 14 (45.2) | 3 (9.7) |
| Anticipated an over burden on NCD services when routine care eventually resumed (n=30) | 24 (77.4) | 6 (19.4) | N/A |
| Had plans for gradual/spaced out resumption of routine care (n=30) | 11 (35.5) | 19 (61.3) | N/A |
| Need to ensure that whole populations have increased health promotion and prevention schemes to decrease vulnerabilities and predispositions to NCDs and optimise prognoses during infectious epidemics | 29 (93.5) | 2 (6.5) | N/A |
| *Use of health apps for chronic NCD care during COVID-19: n (%)* | **Very likely** | **Somewhat likely** | **Not likely** |
| Likely to depend on telehealth, digital health, mobile health applications for follow-ups and NCD care | 7 (22.6) | 18 (58.1) | 6 (19.4) |
| | **Yes** | **No** | **Unsure** |
| Digital health could be an option for NCD patients to experience more autonomy over their care | 16 (51.6) | 4 (12.9) | 11 (35.5) |
| Consider using digital health applications for medical information sharing, especially for patients in multidisciplinary care settings | 27 (87.1) | 2 (6.5) | 2 (6.5) |
| *Personal perceptions of COVID-19: n (%)* | **Very much indeed/ quite a lot** | **Not very much** | **Not at all** |
| Fearful of contracting the virus (whether in contact with COVID-19 patients or not) (n=29) | 16 (51.6) | 11 (35.5) | 2 (6.5) |
| | **Yes** | **No** | **N/A** |
| Fearful of contracting the virus and passing it on to family and friends (n=29) | 23 (74.2) | 3 (9.7) | 3 (9.7) |
| If YES, you isolated yourself due to this fear (n=23) | 8 (34.8) | 15 (65.2) | 0 |
| *Provision of mental healthcare: n (%)* | **Yes** | **No** | **Unsure** |
| Health system had protocols on mental health screening for patients with chronic disease | 4 (12.9) | 16 (54.8) | 11 (41.9) |
| Patients that had contracted SARS-CoV-2 had mental health assessments conducted on them | 1 (3.2) | 17 (54.8) | 13 (41.9) |
| Chronically ill patients that subsequently contracted SARS-CoV-2 had mental health assessments | 1 (3.2) | 17 (54.8) | 13 (41.9) |
| Country had dedicated mental health helpline owing to the COVID-19 pandemic | 7 (22.6) | 12 (38.7) | 12 (38.7) |
| *Ethical issues related to COVID-19* | **Yes** | **No** | **Unsure** |
| Thought it was ethical to obligate to vaccinate against COVID-19 | 10 (32.3) | 19 (61.3) | 2 (6.5) |
| Thought it was ethical to obligate vaccine serology/immune cards/passports to show vaccine/serology/immune status for social activities | 13 (41.9) | 14 (45.2) | 4 (12.9) |
| Thought it was ethical to introduce surveillance measures that adjoin personal details to symptom status (i.e., mobile apps, wristband monitoring) | 6 (19.4) | 19 (61.3) | 6 (19.4) |

**Table 1.** (Continued)

| Provision of COVID-19 care during the pandemic: n (%) | Completely | Quite a bit | A little bit/ not at all |
|---|---|---|---|
| | Most definitely yes | Somewhat yes | A little bit/ Not at all |
| Agreed with measures that obligated physical distancing | 10 (32.3) | 16 (51.6) | 5 (16.1) |
| Thought police/military interventions to ensure physical distancing were done in a fair manner | 6 (29.0) | 17 (54.8) | 5 (16.1) |

*n=31 except where it is indicated; N/A: not applicable

retinopathy, etc. were cancelled. The redirection of services for COVID-19 care further exacerbated the cancellations of elective surgery cases because of the lack of patient transportation services from peripheral hospitals. HPs believed that these cancellations likely had severe consequences for patients with poorer outcomes as well as for healthcare services. Regarding the latter, waiting times for appointments were prolonged for critical surgical care and huge backlogs were further increased in an already overburdened healthcare system.

A PHO reported that there were delays in treatment because of the pandemic, which negatively affected the patient's health in both the private and public sectors, but the situation was much worse in the public sector. For example, oncology patients treated in the public sector waited for many months before commencing chemotherapy or radiotherapy, and those requiring surgery may have been scheduled for surgery up to a year later. There was an urgent need to evaluate and improve such services. Equally, routine community and hospital-based screening and diagnosis for diabetes and hypertension were disrupted, thereby delaying the onset of care for newly diagnosed patients.

A HW reported that "*Covid-19 compromised staffing a lot and that led to wards with patients who had no Covid receiving less attention than patients in Covid wards*."

HPs were negatively affected by the health system's inability to continue caring for patients with NCDs (77%, n=24) (Table 1) with 15 reporting that it bothered them 'very much indeed/ quite a lot' (S2 Fig). Among all HP, 45% worried that NCD patients who had not been followed up due to the crisis would have worsened outcomes (Table 1).

HPs were unhappy about the absence of planning for the provisions of NCD care during the COVID-19 pandemic in South Africa, stating that proactive care for such patients during the crisis was lacking. NCD care had been overlooked by the health minister with no direction or solutions provided to deal with NCD patients during the COVID-19 pandemic.

A HW described the situation as follows: "*I think the pandemic may have played a role in highlighting how broken our health care system is. I hope it has prompted the development of better strategies to deal with crisis in the health system for future pandemics*."

This was reiterated by a PHO who described the desperate situation of those with NCDs seeking emergency healthcare: "*Many times when ambulance services were called out for emergencies like chest pains, CVA* [cerebrovascular accident], *hperglycemia hospitals were full or not accepting cases. Patients were often left to wait for a day or two if sent to causality*."

Even routine appointments were difficult to schedule, as described by a HW: "*A lot of patients struggled to get appointments at doctors offices because they only saw critical patients due to a lot of their time being taken up with extra Covid responsibilities. A lot of elective surgeries were also cancelled and only emergency surgeries were made priority*."

Another HW expressed the following: "*Due to the pandemic, many patients have been unable to or have been hesitant to come to hospitals/clinics for follow-ups on their disease or condition. This resulted in mismanagement as it is difficult to monitor and pick up problems earlier on in a patients care. This is detrimental to some patients as regular monitoring is*

**Table 2. Healthcare workers' perceptions of healthcare provision and patient care during the COVID-19 pandemic[*].**

| Provision of COVID-19 care during the pandemic: n (%) | Completely | Quite a bit | A little bit/ not at all |
|---|---|---|---|
| As a frontline worker, you received adequate personal protection (n=21) | 8 (32.0) | 12 (48.0) | 1 (4.0) |
| | **Yes** | **No** | **Unsure** |
| Adequate provision of individual protection devices (n=23) | 16 (64.0) | 7 (28.0) | N/A |
| Personally worked with COVID-19 patients | 20 (80.0) | 5 (20.0) | N/A |
| Dedicated personnel were often rotated so as not to tire them (n=23) | 14 (56.0) | 10 (40.0) | N/A |
| Felt there was some confusion about how to treat a COVID patient, e.g., frequent changes in drug indications and dosages, protocols (n=11) | 8 (32.0) | 3 (12.0) | N/A |
| Doctors and nurses were placed on COVID-19 patient wards long enough to learn protocols and acquire skills | 8 (32.0) | 2 (8.0) | 15 (60.0) |
| Thought it was ethical to have non-internal medicine physicians treating those critically ill with COVID-19 | 13 (52.0) | 12 (48.0) | N/A |
| Thought it was ethical to have non-internal medicine physicians taken off their service to provide care for COVID-19 patients at the expense of the patients in other specialties | 10 (40.0) | 15 (60.0) | N/A |
| | **Mostly well/ very well** | **Some- times well** | **Not well/Did not use PPE** |
| General perception of healthcare workers by patients whilst wearing personal protection equipment (PPE) | 19 (76.0) | 6 (24.0) | 0 |
| | **Yes** | **No** | **N/A** |
| Felt detached from patients whilst wearing PPE | 11 (44.0) | 13 (52.0) | 1 (4.0) |
| | **Many** | **few** | **None** |
| Proportion of NCD patients abstained from visiting the hospital due to fear of contracting SARS-CoV-2 | 17 (68.0) | 8 (32.0) | 0 |
| | **Very important** | **Important** | **Not important** |
| Importance of adequate support by Palliative Care Specialists to deal with end-stage disease patients (and their families) | 8 (32.0) | 10 (40.0) | 7 (28.0) |
| *Provision of non-communicable disease (NCD) care during the pandemic: n (%)* | **Yes** | **No** | **Unsure** |
| Thought ethical standards of care were upheld for NCD patients in your institution | 12 (48.0) | 5 (20.0) | 8 (32.0) |
| Children's care in your institution was compromised in any way | 5 (20.0) | 15 (60.0) | 5 (20.0) |
| Had concerns about information availability during the pandemic | 16 (64.0) | 9 (36.0) | 0 |
| Traditional delineation of infectious and NCDs should be eliminated to provide holistic care to counter damaging outcomes during infectious epidemics | 14 (56.0) | 6 (24.0) | 5 (20.0) |
| Patients with NCDs expressed anxiety over clinic appointments, follow-up appointments, and general lack of accessibility to physician care | 20 (80.0) | 5 (20.0) | 0 |
| | **Yes** | **No** | **N/A** |
| Had discussions reassuring patients with NCD/ chronic disease about their NCD care | 17 (68.0) | 6 (24.0) | 2 (8.0) |
| Prescribed anxiolytics, anti-depressants, or other psychotropic drugs to patients with NCDs owing to increased anxiety, feelings of depression, stress | 5 (20.0) | 12 (48.0) | 8 (32.0) |
| Patients who would have received a cubicle/isolated space in a department/ward, were placed in shared space/ward beds (n=24) | 15 (60.0) | 9 (36.0) | 0 |

[*]n=25 except where it is indicated.

*important to make sure they are managing their conditions effectively to prevent relapses (e.g., malnutrition in paediatrics/adults, cancer patients, uncontrolled diabetes/hypertension/other chronic diseases of lifestyle."*

From the chronic NCDs, HPs listed diabetes care among the most to be affected (45%, n=14/31) as well as hypertension care (29%, n=9/31) (S3 Fig). Other NCDs whose care was affected during the COVID-19 pandemic, as listed by 17 HPs, included autoimmune diseases,

cancers, chronic obstructive pulmonary disease, dyslipidaemia, epilepsy, heart attack, metabolic syndrome, obesity and renal failure (S3 Fig).

According to HWs, certain measures were implemented for NCD care during the COVID-19 pandemic when services were redirected. Patients who were controlled on their chronic medication were given treatment to cover them for several months as opposed to being seen monthly. Consequently, some patients may have only had a check-up after six months, which was considered problematic. A HW summed up this approach as: "…*most systems fell apart and patients did receive their medication but lacked adequate care.*"

Most HPs (77%) anticipated a greater overburdening on NCD services when routine care eventually resumed (Table 1). Moreover, many (61%) felt that there were no plans for the gradual/spaced out resumption of routine care after the pandemic.

## Ethical issues relating to COVID-19 preventive measures

Most HWs (60%) did not think it was ethical to have non-internal medicine physicians taken off their service to provide care for COVID-19 patients at the expense of the patients in other specialities (Table 2). A similar number thought it was ethical (52%) and unethical (48%) to have non-internal medicine physicians treating those critically ill with COVID-19.

Most HPs (61%) felt that it was unethical to obligate COVID-19 vaccinations, but a similar number felt it was ethical (42%) vs. unethical (45%) to obligate vaccine serology/immune cards/passports to show vaccine/serology/immune status for social activities (Table 1). Most (61%) also felt that it was unethical to introduce surveillance measures that adjoin personal details to symptom status (i.e., mobile apps, wristband monitoring).

Among HPs, 52% agreed 'somewhat' and only 32% agreed 'most definitely' with measures that obligated physical distancing (Table 1). With regards to the enforcement of physical distancing, 55% of HPs did not think that police/military interventions were done in a fair manner (Table 1).

## Utility of digital health in South Africa

HPs were open to the potential use of telehealth, digital health, and mobile health applications with 58% and 23% reporting that they were 'somewhat likely' and 'very likely', respectively, to depend on these for follow-up and NCD care (Table 1). About half (52%) felt that digital health could very likely be an option for NCD patients to experience more autonomy over their care. Most HPs (87%) were very likely to consider using digital health applications for medical information sharing, especially for patients in multidisciplinary care settings.

Some HWs articulated their visions clearly on the potential utility and role of digital technology in healthcare practice. "*Outpatient department; initial telehealth consult to determine severity and whether in patient assessment is required, adjusting medications or add additional medications where needed and continuation of follow-up care.*" (HW)

HWs suggested the use of telehealth to overcome barriers related to physical access of healthcare services. "*Digital health could be used to conduct tele-consultations particularly for patients who are unable to present due to reasons such as disability, to the health facility or are COVID-19+. I believe this will assist patients greatly.*" (HW)

Another HW underlined the utility of digital health during pandemics: "*An accessible online platform must be in place to accommodate the patients during times of isolation.*"

The HW explained that digital health was used during the COVID-19 pandemic where, for example, dieticians consulted patients online when they were too ill to visit the office, did not wish to leave their homes or felt unsafe to enter hospitals.

Although the potential use of telemedicine for patient treatment monitoring and feedback, and for better access for those living long distances from healthcare facilities was highlighted by HPs, some HWs felt that access to digital platforms may be challenging for patients. A HW articulated that *"Many patients attending CHCs for NCD care are older and from disadvantaged communities and do not have smart phones."* Other HWs felt that there was *"Very limited application given in our setting, public patients don't have routine access to digital platforms"* and *"vulnerable patients do not have access to technology."*

## Perceptions and management of anxiety by healthcare professionals during COVID-19

Anxiety, firstly, affected HPs themselves; 48% were fearful of contracting the virus, irrespective of whether they were in contact with COVID-19 patients or not. Further, most (65%) were fearful of contracting the virus and passing it on to family and friends. However, the fear did not cause most of those who were fearful to isolate themselves (70%) (Table 1). Most HWs (64%) had concerns about information availability during the pandemic (Table 2).

Additionally, most HWs (80%) reported that patients with NCDs expressed anxiety over clinic appointments, follow-up appointments, and general lack of accessibility to physician care, and the majority (68%) reassured their patients with NCDs/chronic diseases about their NCD care (Table 2). Nevertheless, most HWs (68%) reported that many NCD patients abstained from visiting hospitals due to fear of contracting SARS-CoV-2. PLWNCDs were afraid of contracting COVID-19 in the healthcare setting as the public health messages delivered indicated they had a heightened risk of infection, poorer outcomes and death thus keeping them away from care.

A PHO described the fear and anxiety expressed by patients: *"Many patients were afraid to go to hospital and clinics to collect medication/see doctor for script renewals due to the fear of contacting Covid 19. Many elders defaulted on chronic medication due to this and the long waiting lines."*

A HW reiterated these sentiments: *"The notion that if people present to hospitals they will contract covid 19 needs to be addressed with the public and removed so that people are not afraid to present themselves for follow-ups and for queries."*

Another HW opined: *"Lot of stigma attached to patients with chronic illness contracting COVID-19, that made them reluctant to seek healthcare during the hard lockdowns."*

While 42% of HP were unsure about the availability of protocols for mental health screening for patients with chronic disease, 55% reported an absence of such protocols (Table 1). Further, 55% reported that mental health assessments were not performed for patients with SARS-CoV-2 nor for chronically ill patients that subsequently contracted SARS-CoV-2. With regards to a dedicated mental health helpline owing to the COVID-19 pandemic, an equal number were unsure about the presence of such a service as those who stated it was not provided (39% for both). Some HWs (20%) prescribed anxiolytics, anti-depressants, or other psychotropic drugs to PLWNCDs who experienced increased anxiety, feelings of depression or stress (Table 2).

2) **Participants with chronic NCDS**

## Participant characteristics

Among the 79 PLWNCDs in the study who had a median age (25th–75th percentiles) of 61.0 (66.0–72.0) years and an age range of 25–90 years, most were women (81%) and received care in the public sector (81%) (Fig 1). The most prevalent NCDs among PLWNCDs were

hypertension (75%), diabetes (47%) and dyslipidaemia (33%). Other NCDs (32%) among PLWNCDs included asthma, cancer, stroke, heart attack, depression, anxiety, eczema, epilepsy, systemic lupus erythematosus and bipolar disorder. There was a similar distribution in the number of NCDs across PLWNCDs; 26 had a single NCD, 24 had two and 26 had ≥3 NCDs. Most PLWNCDs (81%) received care in the public sector with 15% receiving care in the private sector and few (4%) utilising both healthcare sectors.

### NCD care received during COVID-19

Only 51% of PLWNCDs felt that they had received adequate care for their chronic disease during the pandemic 'many times or all the time' (Table 3). In contrast, 27% felt they received 'very little' adequate care or 'sometimes' received adequate care while 23% did not receive adequate care at all. Most PLWNCDs (79%) did not struggle at all to get their routine prescription filled or acquire regular medication during the pandemic, and 82% did not have any routine appointments cancelled. Of those that did have their appointments cancelled (n=14), half (n=7) had a single appointment cancelled and four PLWNCDs had two cancelled (S4 Fig). Among PLWNCDs with cancellations, 36% felt that this may have detrimental effects on their future health.

### Perceptions and mental health status during COVID-19

Among PLWNCDs, 24% felt vulnerable due to their chronic condition during the pandemic 'many times or all the time', and 30% felt vulnerable 'sometimes or very little' (Table 3). Some PLWNCDs (19%) felt hesitant to seek medical attention for non-respiratory medical problems 'many times or all the time', while 72% felt hesitant 'sometimes or very little'. Most (68%) did not struggle to get a medical certificate to stop working.

Among PLWNCDs, 30% were affected by the uncertainty and unknowns about the impact of COVID-19 on chronic NCDs and 25% felt frustrated about the effect of COVID-19 management on their chronic NCDs (Table 3). Some (14%) felt their health status worsened during COVID-19 pandemic. A substantial proportion (43%) were worried about the future state of social interaction.

Many PLWNCDs (52%) felt anxious, lonely or frightened during the pandemic, and 15% felt their mental health deteriorated (Table 3). However, only 15% of those who felt anxious or lonely sought medical attention; of the six PLWNCDs that did, three were prescribed antidepressants or anxiolytics. About a quarter of all PLWNCDs (24%) were encouraged by their healthcare provider to use non-pharmaceutical interventions for any stress or anxiousness experienced.

### Availability of information and the use of health apps during the pandemic

While 65% of PLWNCDs felt informed enough about the danger of COVID-19 linked with chronic NCDs, 29% struggled to find out the veracity of any information circulating (Table 3). A substantial proportion of PLWNCDs (43%) felt affected by the amount of information circulating about COVID-19.

Many PLWNCDs (57%) felt that digital health apps could contribute to better health provision during another crisis like COVID-19 pandemic, but only a third were aware that health apps were offered to help manage the different aspects of COVID-19 in South Africa (Table 3). Among PLWNCDs that were aware (n=26), the majority (n=20) did use health apps to help manage the different aspects of the COVID-19 pandemic. Most PLWNCDs who were aware of the health apps thought the quality of the available health apps was adequate to help manage different aspects of COVID-19 (n=19).

3) **Lessons for the future**

## Improving healthcare delivery during crises in South Africa: HP perspectives

According to HPs, the COVID-19 pandemic highlighted the shortcomings in the South African public healthcare system. The redirecting of resources with a concerted focus on COVID-19 management at the expense of continuity of care for NCDs and other healthcare services led to the neglect of the latter with serious health consequences for patients themselves and the healthcare system. Going forward, HPs felt that NCD care needed to be included in healthcare plans for future pandemics/crises.

The general feeling was that "…*NCD care should not be compromised regardless of pandemic as poor NCD care may worsen the pandemic outcomes*."

A PHO advised that: "*The health care services should put plans in place to ensure the continuation of NCD care and treatment goes uninterrupted should we be faced to another infectious disease outbreak*."

Another HW suggestion was to: "*Incorporate NCD-related objectives to pandemic preparedness plans and ensure sufficient human resources to prevent existing staff from being redirected leaving their core duties unattended too*."

There needed to be sufficient staff attending to NCD care, particularly at the primary healthcare level, to prevent redirection of staff and adverse consequences. The example given was of programme co-ordinators and community health workers being redirected to COVID-19 services, at the expense of other health services; they were unable to fulfil their regular duties.

A PHO stated that to avoid compromises in future NCD care, they had changed their strategy: "*We have broadened disaster management to include natural disasters and pandemics. Changed our scope from a hospital service provider to a healthcare service provider to ensure we have a diversified approach*."

Further, it appears crucial to strengthen referral pathways and ensure that elective surgery continues during crises, particularly for cases with potential complications. This would prevent surgical wards and operating theatres from remaining empty, and avoid very long theatre booking lists post-pandemic, as has occurred locally during and after the COVID-19 pandemic.

A greater focus on health education of both COVID-19 and NCDs for PLWNCDs was a common opinion held among HWs. "*Give prior education to patients to look out for early signs and symptoms. Have more regular check-ups so there is no sudden emergency. Outpatient rehabilitation programmes to manage the condition*." (HW)

This would also improve patient adherence, compliance and co-operation, and may be achieved by community healthcare workers providing contextual community-based care and conducting home visits. Community healthcare workers would need in-depth training on NCD treatment goals, etc. Suggestions from HWs to improve service delivery included better equipping local clinics to provide healthcare services and the decentralisation of care to rural clinics with training of lower-level health workers, e.g., train nurses to deliver renal services, follow-up wound care, etc.

A PHO summarised: "*Promote and encourage primary health care services, promote management of lifestyle diseases efficiently*".

Other suggestions for strengthening the public healthcare system included public-private partnerships. An example provided by a HW was for public sector patients to collect their medications from private healthcare pharmacies. This would lighten the workload in the public healthcare sector and likely reduce patient waiting times.

Additionally, 94% of HPs felt there was a need, at the population level, for increased health promotion and prevention schemes to decrease vulnerabilities and predispositions to NCDs,

**Table 3. Experiences and perceptions of patients with chronic NCDs on their healthcare, mental state and use of health applications during the COVID-19 pandemic[*].**

| Experiences of NCD care during COVID-19 | All the time | Some-times | Not at all |
|---|---|---|---|
| Felt adequate care was received for chronic disease during the pandemic | 40 (50.6) | 21 (26.6) | 18 (22.8) |
| Struggled to get routine prescription filled or acquire regular medication since March 2020 | 7 (8.9) | 10 (12.7) | 62 (78.5) |
| | **Yes** | **No** | **N/A** |
| Any routine appointments for your NCD were cancelled due to COVID-19 | 14 (17.7) | 65 (82.3) | – |
| Struggled to get medical certificate to stop working | 2 (2.5) | 23 (29.1) | 54 (68.4) |
| *Perceptions and mental health status during COVID-19* | **Many times/all the time** | **Very little** | **Not at all** |
| Felt vulnerable due to your chronic condition during the pandemic | 19 (24.1) | 24 (30.4) | 36 (45.6) |
| | **Yes** | **No** | **Unsure** |
| Felt hesitant to seek medical attention for non-respiratory medical problems | 15 (19.0) | 57 (72.2) | 7 (8.9) |
| Affected by the uncertainty and unknowns about the impact of COVID-19 on your NCDs | 24 (30.4) | 45 (57.0) | 10 (12.7) |
| Felt frustrated about the management of COVID-19 on chronic disease | 20 (25.3) | 51 (64.6) | 8 (10.1) |
| Felt may be at risk even after the COVID-19 threat has settled | 35 (44.3) | 32 (40.5) | 12 (15.2) |
| Worried about the future state of social interaction | 34 (43.0) | 39 (49.4) | 6 (7.6) |
| Felt your health status worsened during COVID-19 pandemic | 14 (17.7) | 57 (72.2) | 8 (10.1) |
| Felt mental health deterioration | 12 (15.2) | 59 (74.7) | 8 (10.1) |
| Felt anxious/lonely/frightened | 41 (51.9) | 38 (48.1) | – |
| | **Yes** | **No** | **N/A** |
| Sought medical attention for feelings of loneliness or anxiety (n=40) | 6 (15.0) | 34 (85.0) | – |
| If YES, you were prescribed anti-depressants (n=5) | 3 (60.0) | 2 (40.0) | – |
| Healthcare provider encouraged non-pharmaceutical interventions for any stress or anxiousness experienced | 19 (24.1) | 37 (46.8) | 23 (29.1) |
| *Availability of information and the use of health apps during the pandemic* | **Yes** | **No** | **Unsure** |
| Felt affected by the amount of information circulating about the disease | 34 (43.0) | 39 (49.4) | 6 (7.6) |
| Struggled to find out the veracity of any information circulating | 23 (29.1) | 43 (54.4) | 13 (16.5) |
| Felt informed enough about the danger of COVID-19 linked with your NCD | 51 (64.6) | 19 (24.1) | 9 (11.4) |
| Health apps were offered to help you manage the different aspects of COVID-19 | 26 (32.9) | 43 (54.4) | 10 (12.7) |
| If YES, you used health apps to help you manage the different aspects of COVID-19 (n=26) | 20 (76.9) | 5 (19.2) | 1 (3.8) |
| If YES, thought the quality of available health apps was adequate to help manage different aspects of COVID-19 (n=26) | 19 (73.1) | 2 (7.7) | 5 (19.2) |

*(Continued)*

**Table 3.** (Continued)

| Experiences of NCD care during COVID-19 | All the time | Some-times | Not at all |
|---|---|---|---|
| Digital health apps could improve chances of better health provided during another crisis like COVID-19 pandemic | 45 (57.0) | 5 (6.3) | 29 (36.7) |

*n=79 except where it is indicated; NCDs: non-communicable diseases; N/A: not applicable.

and to optimise prognoses during infectious epidemics (Table 1). Over half the HWs (56%) felt that traditional delineation of infectious and NCDs should be eliminated to provide holistic care to counter damaging outcomes during infectious epidemics (Table 2).

The sentiments expressed on the effects of the pandemic on NCD care and prevention medicine were similar across HPs and can be summarised as follows: "*COVID had a big impact on the treating of other conditions. Health educations needs to be focused around the NCD conditions.*" (PHO)

However, diverse views were expressed on the ability of the healthcare system to reform. Some felt that NCD care would be further neglected, and "*…our healthcare system will never learn from its mistakes.*". (HW)

Others were more optimistic: "*I think we are better equiped and knowledgeable with the pandemic and can improve on the quality of care and management we render to NCD patients. Yes, when the pandemic started NCD patients were somewhat "neglected" because all attention was given to this new deadly disease (Covid-19). Now we know and have seen better.*" (HW)

## Discussion

This study highlights the dire state of the South African healthcare system with a perception among HPs of poor planning, and a lack of policies and strategies to care for non-COVID-19 illnesses during the pandemic. HPs perceived that many patients, who required urgent medical care and could have been easily treated, may have complicated or succumbed to their illness because of the lack of care. NCD care was neglected with subsequent serious consequences for patients and the healthcare system. The lack of continuity of care for PLWNCDs was perceived to lead to greater complications. This was supported by findings from Cape Town where NCD service disruptions contributed to marked decreases in the number of HbA1c tests conducted while the proportion of patients with uncontrolled diabetes increased [17].

The sentiments of poor NCD care expressed in this study are in keeping with the findings from a WHO survey which reported severe disruptions in prevention and treatment services for NCDs during the COVID-19 pandemic [4]. Rehabilitation services, diabetes and diabetes complications management, and hypertension care were the three main services to be affected. The perceptions of HPs in this study were echoed by those in a global survey of nine countries including eight LMICs [5]. HPs felt that the key factors influencing patient care included patient reluctance to seek care during the pandemic (on account of their anxiety of contracting COVID-19 in healthcare facilities), delays in care for non-COVID-19 illnesses and the loss of continuity of care. This underscores the need for a contingency plan that not only addresses the crisis, but also allows for the continuity of services and care for non-crisis related illnesses. Such strategies are crucial to prevent unnecessary morbidity and mortality. There needs to be multiple clear pathways for the dissemination of information to the public with regards to access and availability of healthcare services as well as education to reassure patients of protection from the crisis if clear protocols are followed.

The South African Department of Health instituted a protocol through the Central Chronic Medicine Dispensing and Distribution Programme (CCMMD) to allow stable patients to be issued with more medication to ensure longer return date and avoid the unnecessary risk of visiting the hospital. Nevertheless, a community-based follow-up mechanism such as an accompanying digital health solution to monitor and support PLWNCDs was needed to ensure their care was not compromised.

Perceptions of inadequate care were repeated by PLWNCDs in this study. Although most PLWNCDs did not struggle to acquire their regular medication during the pandemic, nor did they have their appointments cancelled, 50% of PLWNCDs reported that they had received 'very little' adequate care, 'sometimes' received adequate care or did not receive adequate care at all. Moreover, while a majority of PLWNCDs felt anxious, lonely or frightened during the pandemic, only a small proportion sought medical attention. Poor mental health is a growing burden in South Africa but generally receives little attention at the primary healthcare level. This occurs despite over a third of patients with hypertension or diabetes in the country having symptoms of depression and anxiety [18], with this likely worsening during the pandemic. Public health measures, such as the hard lockdown in South Africa, disrupted social structures. These included family visits, social gatherings, church attendances and in-person working which are platforms that dissipate community anxieties and assist with coping. Adverse mental health such as feelings of being anxious, lonely or frightened during the pandemic as well as the uncertainty and frustration about the impact of COVID-19 on their NCDs described by PLWNCDs in this study accord with the literature [6,9,19]. Deterioration in self-perceived mental health was a global phenomenon as reported in an international survey conducted in 65 countries [9].

The need for physical distancing and isolation during the pandemic underlined the utility of digital health, which was generally positively perceived by both HP and PLWNCDs in this study. This accords with the global trend; indeed, in India, a mobile-based application introduced in 2018, was used to track the follow-up of PLWNCDs and connect those requiring healthcare services with providers during the pandemic [9,20]. Further research is needed in South Africa to incorporate the use of digital health apps in routine NCD care to enable better service delivery during times of crises as well as to ease the burden on the currently strained healthcare system. Such research will need to consider the barriers to the uptake of digital health in South Africa including disruptions in power supply, inability to access technology, poor literacy levels, etc.

## Strengths and limitations

The inclusion of both HPs and PLWNCDs is a major strength of this study as it enabled the perceptions of both healthcare providers and patients to be examined. However, the small sample size is a limitation and prevents generalisability to other settings. The inability to target the minimum 10 participants for PHOs within the study timeframe was a limitation, which may or may not have affected the study findings. None of the PHOs included in this study worked in the public healthcare sector; this may have affected the study findings because the majority of South Africans utilise public healthcare facilities, and these services were likely more overburdened than those in the private sector. While the convenience snowball sampling strategy used may have resulted in a potential sampling bias where participants included may have had similar traits or experiences, it was the most feasible sampling approach. This recruitment strategy may have contributed to the older age of PLWNCDs in this study and is therefore a limitation. The use of open-ended written questions for the qualitative data collection versus conducting in-depth interviews is a limitation because it precluded a deeper

exploration of the relevant issues. Since there were no prior services dedicated to COVID-19, the communities of practice and network referrals used ensured that the most appropriate HP involved in COVID-19 care were interviewed. These study findings are not generalisable to all PLWNCDs in South Africa as this study did not include a representative sample of PLWNCDs with debilitating NCD related complications such as stroke, retinopathy, nephropathy, etc. Such PLWNCDs are likely to require more specialised care and frequent contact with healthcare services, which they may not have been able to access during the COVID-19 pandemic. Similarly, the findings may not be generalisable to all HPs who may have had different experiences depending on their specialisation or the facility where they worked.

## Conclusions

This study demonstrates that the COVID-19 pandemic had a severe impact in South Africa and negatively affected its HPs and PLWNCDs. Moreover, the pandemic exposed and worsened the pre-pandemic shortcomings in the healthcare system where NCDs have less policy attention and resources. The South African healthcare system was unable to provide optimal care for and protect PLWNCDs during the COVID-19 pandemic, particularly those who needed emergency and specialised NCD care. Strategies and policies are needed to ensure that NCD care and continuity of its essential service components is part of the public health response in emergencies. This will ensure that NCDs and Mental Health services are not neglected during future crises and that PLWNCDs continue to access healthcare services timeously during such periods. Systematic approaches to digital health care solutions may be developed to assist healthcare services to be better equipped in future [4]. Such strategies may utilise digital health apps to improve care and to address the mental healthcare needs of PLWNCDs. The lack of any documented impact of suboptimal NCD and Mental Health care during the pandemic warrants further research on the indirect impact of the COVID-19 public health response. The latter focused mostly on the acute crisis while inadvertently neglecting chronic diseases care.

## Supporting information

**S1 Data.** XXX.
(XLSX)

**Supplementary Fig 1. Characteristics of healthcare professionals (n=31)** .
(DOCX)

**Supplementary Fig 2. The degree to which healthcare professionals were negatively affected by the health system's inability to continue caring for patients with NCDs (n=21)** .
(DOCX)

**Supplementary Fig 3. Care of chronic NCDs listed by healthcare professionals (n=17) to be most affected during the COVID-19 pandemic** .
(DOCX)

**Supplementary Fig 4. Cancellation of appointments and perceptions of this on future health** .
(DOCX)

## Author contributions

**Conceptualization:** Nasheeta Peer, Michel Oris, Kibachio Mwangi, Sugitha Sureshkumar, Andre-Pascal Kengne.

**Data curation:** Nasheeta Peer, Tshephang Mashiane.

**Formal analysis:** Nasheeta Peer, Tshephang Mashiane.

**Investigation:** Tshephang Mashiane.

**Methodology:** Nasheeta Peer, Andre-Pascal Kengne.

**Project administration:** Tshephang Mashiane.

**Supervision:** Nasheeta Peer.

**Writing – original draft:** Nasheeta Peer.

**Writing – review & editing:** Tshephang Mashiane, Michel Oris, Kibachio Mwangi, Sugitha Sureshkumar, Andre-Pascal Kengne.

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
