## [Decision Letter · Decision Letter 0]

3 Jan 2024

PONE-D-23-32125Non-communicable disease and mental health care during the COVID-19 pandemic in South Africa: perspectives from healthcare professionals and patientsPLOS ONE

Dear Dr. Peer,

Thank you for submitting your manuscript to PLOS ONE. After careful consideration, we feel that it has merit but does not fully meet PLOS ONE’s publication criteria as it currently stands. Therefore, we invite you to submit a revised version of the manuscript that addresses the points raised during the review process.

We look forward to receiving your revised manuscript.

Kind regards,

Yogan Pillay, Phd

Academic Editor

PLOS ONE

“This study was funded by the SAMRC through baseline allocation to the Non-Communicable Diseases Research Unit (NCDRU).”

3. In the online submission form you indicate that your data is not available for proprietary reasons and have provided a contact point for accessing this data. Please note that your current contact point is a co-author on this manuscript. According to our Data Policy, the contact point must not be an author on the manuscript and must be an institutional contact, ideally not an individual. Please revise your data statement to a non-author institutional point of contact, such as a data access or ethics committee, and send this to us via return email. Please also include contact information for the third party organization, and please include the full citation of where the data can be found.

Additional Editor Comments:

As above

Reviewers' comments:

Reviewer's Responses to Questions

**Comments to the Author**

1. Is the manuscript technically sound, and do the data support the conclusions?

Reviewer #1: Yes

2. Has the statistical analysis been performed appropriately and rigorously? 

Reviewer #1: Yes

3. Have the authors made all data underlying the findings in their manuscript fully available?

Reviewer #1: Yes

4. Is the manuscript presented in an intelligible fashion and written in standard English?

Reviewer #1: Yes

5. Review Comments to the Author

Reviewer #1: Non-communicable disease and mental health care during the COVID-19 pandemic in South Africa: perspectives from healthcare professionals and patients

The manuscript reflects an important area of study, however there some concerns at a methodological level. The sample size is rather small for the quantitative arm of the research, and the extent to which the findings can be taken nationally representative is questionable. The qualitative aspect of the study fails to provide the theoretical background. For example, the analytical procedures are glossed over, without situating it – the analysis seems to use a content analysis approach and should have been articulated as such, with appropriate references. A few other concerns are noted below:

(1) This sweeping sentence (while possible true) needs a reference, otherwise it is conjecture: “Consequently, non-communicable disease (NCD) care for hypertension, diabetes, and other NCDs have not been prioritised with healthcare stakeholders perceiving these conditions to be less important.

(2) Remove ‘both” in this phrase “…both the health, societal and economic levels,…”

(3) This rationale does not make sense, because the recall bias applies to all of the study groups, not just the PHO’s: “the targeted minimum 10 participants per group was not obtained for PHOs. This was because the COVID-19 pandemic had passed its peak and all COVID-19 restrictions (e.g., wearing of face mask, social distancing, temperature checks, curfews, etc.) had been lifted in June 2022 in South Africa, and interviews conducted thereafter may be prone to recall bias.”

(4) Having 6 PHO’s from the private section and none from the public sector is a serious sampling bias, especially considering the vast majority of the population access the public sector facilities which were substantially more stretched.

(5) This is poorly written, suggesting an inadequate understanding of what defines the terms “clinician”: “…HWs included 13 clinicians and 12 other HWs consisting of nurses, pharmacists…” This should be corrected elsewhere in the manuscript as well. See Figure 1 as well.

(6) Check if this is correct: “…median number of years of work experience was 9.0 (4.0-9.0) years…”

(7) Do the authors mean ‘…were affected…” in this sentence “HPs were afflicted by the health system's inability to continue caring for patients…”

(8) This should read ‘seen’: “….as opposed to being seeing monthly…”

(9) The term ‘participants is used in certain places without clarity of which group is being referred to since there were difference participant groups in the study. See, for example “Participants were open to the potential use of telehealth, digital health, mobile health applications with 58% and 23%...”

(10) The findings are discussion relating to the proposed use of digital health technologies completely ignores the issue of access, literacy, network availability (unreliability), electricity unreliability, etc. In addition, what about the fact that most state hospitals do not have functional network systems – this is a very serious problem – I work in one such facility. It is surprising that this was not raised by participants, or queried by the researchers.

(11) This his seems to imply that prescribing medication is the mainstay approach to the treating depression, anxiety and 'stress': “Only 20% of HWs prescribed anxiolytics, anti-depressants, or other psychotropic drugs to PLWNCDs who experienced increased anxiety, feelings of depression or stress.” There should have been some questioning / prompt about psychological therapies offered.

(12) The NCD sample is expected to be a bit older, but the range between 66-72 seems very elderly to be considered an unbiased sampling strategy. This must be explained.

(13) This finding (these responses) should have been enquired into regarding whether the others were offered or referred for psychological interventions: “…of the six participants that did, three were prescribed antidepressants or anxiolytics.”

(14) “ About a quarter of all PLWNCDs (24%)were encouraged by their healthcare provider to use non-pharmaceutical interventions for any stress or anxiousness experienced” Like?? And were they informed were these could be accessed? This should have been queried.

(15) Correct spelling of ‘pandemic’ in one of the excerpts (page 16).

(16) “…only a small proportion sought medical attention, and even fewer were prescribed antidepressants or relaxants.” See my earlier comment. The authors are making more of prescription drugs for common mental disorders (depression & anxiety) than the empirical evidence suggests in terms of treatment efficacy.

(17) This sentence needs improved construction: “Public health measures, such as the hard lockdown in South Africa, disrupted social structures like family visits, social gatherings, church attendances as well as in-person working; these are platforms that dissipate community anxieties and assist with coping.”

(18) Line 4 on page 21 – rather say 'physical distancing' because 'social distancing' was a misnomer used by governments and had more negative effects than commonly realised.

(19) Not clear what this means: “ As the opportunity cost interims of deaths…”

Overall, I think this study could have been better handled methodologically, but Given that there is not much else done in the area, it may be worth considering for publication, subject to the revision along the lines described above.

6. PLOS authors have the option to publish the peer review history of their article (what does this mean? ). If published, this will include your full peer review and any attached files.

**Do you want your identity to be public for this peer review?** For information about this choice, including consent withdrawal, please see our Privacy Policy .

Reviewer #1: No

---

## [Author Response · Author response to Decision Letter 1]

19 Feb 2024

PONE-D-23-32125

Non-communicable disease and mental health care during the COVID-19 pandemic in South Africa: perspectives from healthcare professionals and patients

Dear Editor

We are pleased to resubmit our revised manuscript. We thank the reviewer for his/her helpful comments. The recommendations suggested have substantially improved the manuscript. We trust that the paper will now be acceptable for publication in PLOS ONE.

Please see below for the detailed responses to the reviewer’s comments. The revisions have been included in the manuscript in the track and change format.

Yours sincerely

Prof Peer and colleagues

The manuscript has been formatted in line with the journal requirements. We hope this is now in order.

“This study was funded by the SAMRC through baseline allocation to the Non-Communicable Diseases Research Unit (NCDRU).”

Please include the funding statement as follows: This study was funded by the SAMRC through baseline allocation to the Non-Communicable Diseases Research Unit (NCDRU).

3. In the online submission form you indicate that your data is not available for proprietary reasons and have provided a contact point for accessing this data. Please note that your current contact point is a co-author on this manuscript. According to our Data Policy, the contact point must not be an author on the manuscript and must be an institutional contact, ideally not an individual. Please revise your data statement to a non-author institutional point of contact, such as a data access or ethics committee, and send this to us via return email. Please also include contact information for the third party organization, and please include the full citation of where the data can be found. We have included a link to the dataset: https://medat.samrc.ac.za/index.php/catalog/53

The phrase “data not shown” has been removed and all the relevant data included within the paper.

This has been corrected.

5. Review Comments to the Author

Reviewer #1: Non-communicable disease and mental health care during the COVID-19 pandemic in South Africa: perspectives from healthcare professionals and patients

The manuscript reflects an important area of study, however there some concerns at a methodological level. The sample size is rather small for the quantitative arm of the research, and the extent to which the findings can be taken nationally representative is questionable.

We thank the reviewer for this comment, which we agree with. We have included the following under ‘Strengths and limitations’ (Page 22): “However, the small sample size is a limitation and prevents generalisability to other settings.”

Further limitations re sample size have been included in the manuscript under ‘Strengths and limitations’ as follows (Pages 22-23): “The inability to target the minimum 10 participants for PHOs within the study timeframe was a limitation… These study findings are not generalisable to all PLWNCDs in South Africa… Similarly, the findings may not be generalisable to all HPs…”. We hope this is acceptable.

The qualitative aspect of the study fails to provide the theoretical background. For example, the analytical procedures are glossed over, without situating it – the analysis seems to use a content analysis approach and should have been articulated as such, with appropriate references.

We apologise for this oversight. We have now included a paragraph on the qualitative analyses on Page 7. “The method used to evaluate the open-ended questions was content analysis (14). Valid inferences were systematically and objectively drawn from the HPs’ written text responses to these open-ended questions. Manifest analysis was used in this study with frequent referrals to the original participant text thereby retaining the original meanings and contexts. Inductive reasoning was used to develop the conclusions from the data collected.”

We have also included a limitation of the methodology used on Page 22: “The use of open-ended written questions for the qualitative data collection versus conducting in-depth interviews is a limitation because it precluded a deeper exploration of the relevant issues.”

A few other concerns are noted below:

(1) This sweeping sentence (while possible true) needs a reference, otherwise it is conjecture: “Consequently, non-communicable disease (NCD) care for hypertension, diabetes, and other NCDs have not been prioritised with healthcare stakeholders perceiving these conditions to be less important.

The reference has been included.

(2) Remove ‘both” in this phrase “…both the health, societal and economic levels,…”

We thank the reviewer for picking up this error. The word ‘both’ has been removed.

(3) This rationale does not make sense, because the recall bias applies to all of the study groups, not just the PHO’s: “the targeted minimum 10 participants per group was not obtained for PHOs. This was because the COVID-19 pandemic had passed its peak and all COVID-19 restrictions (e.g., wearing of face mask, social distancing, temperature checks, curfews, etc.) had been lifted in June 2022 in South Africa, and interviews conducted thereafter may be prone to recall bias.”

We agree with the reviewer that recall bias is applicable to all participants. We have amended the text for clarity as follows, and we hope that it now reads better (Pages 6-7): “..the targeted minimum 10 participants per group had not been obtained for PHOs; however, participant numbers for HWs and PLWNCDs had exceeded the minimum required…The fieldwork was terminated at this stage …The study team felt that any interviews conducted after the lifting of restrictions may be prone to greater recall bias (14, 15).” We hope this now reads better.

(4) Having 6 PHO’s from the private section and none from the public sector is a serious sampling bias, especially considering the vast majority of the population access the public sector facilities which were substantially more stretched.

We thank the reviewer for this comment and have added the following under the study limitations (Page 22): “None of the PHOs included in this study worked in the public healthcare sector; this may have affected the study findings because the majority of South Africans utilise public healthcare facilities, and these services were likely more overburdened than those in the private sector.”

(5) This is poorly written, suggesting an inadequate understanding of what defines the terms “clinician”: “…HWs included 13 clinicians and 12 other HWs consisting of nurses, pharmacists…” This should be corrected elsewhere in the manuscript as well. See Figure 1 as well.

This has been reworded as follows (Page 8): “HWs included 13 doctors and 12 other HWs consisting of nurses…”. Figure 1 (Page 27) has also been corrected.

(6) Check if this is correct: “…median number of years of work experience was 9.0 (4.0-9.0) years…”

Thank you for this comment; however, the data are correct. There were 13/31 (42%) HPs with 9 years of work experience, which contributed to the data being very skewed.

(7) Do the authors mean ‘…were affected…” in this sentence “HPs were afflicted by the health system's inability to continue caring for patients…”

Yes, we mean ‘affected’ and have changed the text to ‘negatively affected’.

(8) This should read ‘seen’: “….as opposed to being seeing monthly…”

We thank the reviewer for picking up this error. It has been corrected.

(9) The term ‘participants is used in certain places without clarity of which group is being referred to since there were difference participant groups in the study. See, for example “Participants were open to the potential use of telehealth, digital health, mobile health applications with 58% and 23%...”

This has been corrected throughout the paper where specific participant groups e.g., HPs, PLWNCDs were referred to. We hope the text is now clear.

(10) The findings are discussion relating to the proposed use of digital health technologies completely ignores the issue of access, literacy, network availability (unreliability), electricity unreliability, etc. In addition, what about the fact that most state hospitals do not have functional network systems – this is a very serious problem – I work in one such facility. It is surprising that this was not raised by participants, or queried by the researchers.

Although barriers to the potential provision of digital health care were not included in the study questionnaire, some HWs did express their views on this topic. They felt that access to digital platforms may be challenging for patients. We have expanded on this and included some quotes on Page 13.

Further, we have alluded to the problems/challenges highlighted by the reviewer above on Page 22: “Such research will need to consider the barriers to the uptake of digital health in South Africa including disruptions in power supply, inability to access technology, poor literacy levels, etc.” We hope this is acceptable.

(11) This his seems to imply that prescribing medication is the mainstay approach to the treating depression, anxiety and 'stress': “Only 20% of HWs prescribed anxiolytics, anti-depressants, or other psychotropic drugs to PLWNCDs who experienced increased anxiety, feelings of depression or stress.” There should have been some questioning / prompt about psychological therapies offered.

We agree and thank the reviewer for this comment. We have rephrased the sentence as follows (Page 15): “Some HWs (20%) prescribed anxiolytics, anti-depressants…”.

Although we did not probe HWs further about psychological therapies offered, on Page 16 we note that: “About a quarter of all PLWNCDs (24%) were encouraged by their healthcare provider to use non-pharmaceutical interventions for any stress or anxiousness experienced.”

(12) The NCD sample is expected to be a bit older, but the range between 66-72 seems very elderly to be considered an unbiased sampling strategy. This must be explained.

The median age (25th–75th percentiles) of PLWNCDs was 61.0 (66.0-72.0) years while age range was 25-90 years (range included on Page 15). We agree that this was an older sample with people who were retired/pensioners, etc. more accessible in the community and more agreeable to participate in the study. We have included the older age of participants as a study limitation (Page 22): “This recruitment strategy may have contributed to the older age of PLWNCDs in this study and is therefore a limitation.”

(13) This finding (these responses) should have been enquired into regarding whether the others were offered or referred for psychological interventions: “…of the six participants that did, three were prescribed antidepressants or anxiolytics.”

Although a specific follow-up question was not asked, all PLWNCDs were asked whether “Healthcare provider encouraged non-pharmaceutical interventions for any stress or anxiousness experienced”.

This is reported in Table 3 and on Page 16: “About a quarter of all PLWNCDs (24%) were encouraged by their healthcare provider to use non-pharmaceutical interventions for any stress or anxiousness experienced.”

(14) “ About a quarter of all PLWNCDs (24%)were encouraged by their healthcare provider to use non-pharmaceutical interventions for any stress or anxiousness experienced” Like?? And were they informed were these could be accessed? This should have been queried.

Unfortunately, these details were not expanded on in the study.

(15) Correct spelling of ‘pandemic’ in one of the excerpts (page 16).

This has been corrected, thank you.

(16) “…only a small proportion sought medical attention, and even fewer were prescribed antidepressants or relaxants.” See my earlier comment. The authors are making more of prescription drugs for common mental disorders (depression & anxiety) than the empirical evidence suggests in terms of treatment efficacy.

We thank the reviewer for this comment. The following has been removed: “…and even fewer were prescribed antidepressants or relaxants”.

(17) This sentence needs improved construction: “Public health measures, such as the hard lockdown in South Africa, disrupted social structures like family visits, social gatherings, church attendances as well as in-person working; these are platforms that dissipate community anxieties and assist with coping.”

This has been divided into 2 sentences and we hope it reads better now: “Public health measures, such as the hard lockdown in South Africa, disrupted social structures. These included family visits, social gatherings, church attendances and in-person working which are platforms that dissipate community anxieties and assist with coping.” (Page 21)

(18) Line 4 on page 21 – rather say 'physical distancing' because 'social distancing' was a misnomer used by governments and had more negative effects than commonly realised.

This has been amended, thanks.

(19) Not clear what this means: “ As the opportunity cost interims of deaths…”

We apologise for the lack of clarity. We have divided this into 2 sentences and shortened the text. We hope it is now easier to understand. “The lack of any documented impact of the suboptimal NCD and Mental Health care during the pandemic warrants further research on the indirect impact of the COVID-19 public health response. The latter focused mostly on the acute crisis while inadvertently neglecting chronic diseases care.” (Page 23)

Overall, I think this study could have been better handled methodologically, but Given that there is not much else done in the area, it may be worth considering for publication, subject to the revision along the lines described above.

We thank the reviewer for his/her feedback and hope that our manuscript reads better now.

---

## [Decision Letter · Decision Letter 1]

24 Sep 2024

PONE-D-23-32125R1Non-communicable disease and mental health care during the COVID-19 pandemic in South Africa: perspectives from healthcare professionals and patientsPLOS ONE

Dear Dr. Peer,

Thank you for submitting your manuscript to PLOS ONE. After careful consideration, we feel that it has merit but does not fully meet PLOS ONE’s publication criteria as it currently stands. Therefore, we invite you to submit a revised version of the manuscript that addresses the points raised during the review process.

Thank you for revising your manuscript. Whilst the majority of the concerns have been addressed, per Reviewer 2's assessment, please do take care to ensure that neither your title nor your conclusions overstate the generalizability of your study. In addition, to ensure that your manuscript complies with our requirements for reproducibility and data availability, I would be grateful if you could please address the following points:

Are you able to provide any additional details about the development of the questionnaires, including any pilot testing?Please provide a copy of the questionnaires as a supporting information fileI noted a few references to data not shown. Can the specific data that support these statements be provided as a supplement?

We look forward to receiving your revised manuscript.

Kind regards,

Marianne Clemence

Staff Editor

PLOS ONE

Reviewers' comments:

Reviewer's Responses to Questions

**Comments to the Author**

1. If the authors have adequately addressed your comments raised in a previous round of review and you feel that this manuscript is now acceptable for publication, you may indicate that here to bypass the “Comments to the Author” section, enter your conflict of interest statement in the “Confidential to Editor” section, and submit your "Accept" recommendation.

Reviewer #1: All comments have been addressed

Reviewer #2: All comments have been addressed

2. Is the manuscript technically sound, and do the data support the conclusions?

Reviewer #1: Yes

Reviewer #2: Partly

3. Has the statistical analysis been performed appropriately and rigorously? 

Reviewer #1: Yes

Reviewer #2: Yes

4. Have the authors made all data underlying the findings in their manuscript fully available?

Reviewer #1: Yes

Reviewer #2: Yes

5. Is the manuscript presented in an intelligible fashion and written in standard English?

Reviewer #1: Yes

Reviewer #2: Yes

6. Review Comments to the Author

Reviewer #1: The manuscript can be accepted .....................................................................................................................................................................................................................................................................................................................................................................................................................................................................................................................................................................................................................................................................

Reviewer #2: Due to the small sample size and the use of private sector opinions in this study, it is suggested to modify the title of the article. The authors have corrected all the problems suggested by the first reviewer and have included them in the limitations of the study, but this study cannot be generalized to the whole country and it is better to include phrases such as "in the private sector" or "in a number of clinics" in the title of the article.

7. PLOS authors have the option to publish the peer review history of their article (what does this mean? ). If published, this will include your full peer review and any attached files.

**Do you want your identity to be public for this peer review?** For information about this choice, including consent withdrawal, please see our Privacy Policy .

Reviewer #1: No

Reviewer #2: No

---

## [Author Response · Author response to Decision Letter 2]

4 Oct 2024

PONE-D-23-32125

Non-communicable disease and mental health care during the COVID-19 pandemic in South Africa: perspectives from healthcare professionals and patients

Dear Editor

We are pleased to resubmit our revised manuscript. We thank the reviewer for his/her comments. We hope that the paper will now be acceptable for publication in PLOS ONE.

Please see below for the detailed responses to the Reviewer’s and Editor’s comments. The revisions have been included in the manuscript in the track and change format.

Yours sincerely

Prof Peer and colleagues

Thank you for revising your manuscript. Whilst the majority of the concerns have been addressed, per Reviewer 2's assessment, please do take care to ensure that neither your title nor your conclusions overstate the generalizability of your study. In addition, to ensure that your manuscript complies with our requirements for reproducibility and data availability, I would be grateful if you could please address the following points:

• Are you able to provide any additional details about the development of the questionnaires, including any pilot testing?

We have included the following text on Page 6: “The study questionnaire was developed through consultation with the COVID-19 ELEPHANT study group. The questions were tested for acceptability by two independent groups with diverse professional backgrounds, which further informed the development of the questionnaire.”

• Please provide a copy of the questionnaires as a supporting information file

The study questionnaires have been included.

• I noted a few references to data not shown. Can the specific data that support these statements be provided as a supplement?

We have included the data not shown previously in Supplementary Figures 1-4. We trust this is satisfactory.

Reviewer #2: Due to the small sample size and the use of private sector opinions in this study, it is suggested to modify the title of the article. The authors have corrected all the problems suggested by the first reviewer and have included them in the limitations of the study, but this study cannot be generalized to the whole country and it is better to include phrases such as "in the private sector" or "in a number of clinics" in the title of the article.

We have amended the title to include the word ‘selected’ and it now reads as follows: “Non-communicable disease and mental health care during the COVID-19 pandemic in South Africa: perspectives from selected healthcare professionals and patients”.

We could not include the suggestions made by the reviewer because the healthcare professionals were from both the public (n=13) and private (n=18) sectors. We could not include “clinics” because many health professionals worked in hospitals. We trust this is acceptable.

---

## [Decision Letter · Decision Letter 2]

13 Jan 2025

Non-communicable disease and mental health care during the COVID-19 pandemic in South Africa: perspectives from selected healthcare professionals and patients

PONE-D-23-32125R2

<h3>Dear Dr. Nasheeta Peer,</h3>

We’re pleased to inform you that your manuscript has been judged scientifically suitable for publication and will be formally accepted for publication once it meets all outstanding technical requirements.

Kind regards,

Desalew Tilahun

Academic Editor

PLOS ONE

Additional Editor Comments (optional):

Reviewers' comments:

Reviewer's Responses to Questions

**Comments to the Author**

1. If the authors have adequately addressed your comments raised in a previous round of review and you feel that this manuscript is now acceptable for publication, you may indicate that here to bypass the “Comments to the Author” section, enter your conflict of interest statement in the “Confidential to Editor” section, and submit your "Accept" recommendation.

Reviewer #2: All comments have been addressed

2. Is the manuscript technically sound, and do the data support the conclusions?

Reviewer #2: Yes

3. Has the statistical analysis been performed appropriately and rigorously? 

Reviewer #2: Yes

4. Have the authors made all data underlying the findings in their manuscript fully available?

Reviewer #2: Yes

5. Is the manuscript presented in an intelligible fashion and written in standard English?

Reviewer #2: Yes

6. Review Comments to the Author

Reviewer #2: According to the comments of the first reviewer of the article and the explanations of the authors, in my opinion, the publication of the article is unimpeded. Wishing success to the authors of this article, I hope that their next studies will be carried out in a wider scope in a way that reflects the situation of their country.

7. PLOS authors have the option to publish the peer review history of their article (what does this mean? ). If published, this will include your full peer review and any attached files.

**Do you want your identity to be public for this peer review?** For information about this choice, including consent withdrawal, please see our Privacy Policy .

Reviewer #2: No

---

## [Editor Report · Acceptance letter]

PONE-D-23-32125R2

PLOS ONE

Dear Dr. Peer,

I'm pleased to inform you that your manuscript has been deemed suitable for publication in PLOS ONE. Congratulations! Your manuscript is now being handed over to our production team.

Kind regards,

on behalf of

Mr. Desalew Tilahun Beyene

Academic Editor

PLOS ONE